# Surface Modifications of Titanium Aluminium Vanadium Improve Biocompatibility and Osteogenic Differentiation Potential

**DOI:** 10.3390/ma14061574

**Published:** 2021-03-23

**Authors:** Birgit Lohberger, Nicole Eck, Dietmar Glaenzer, Heike Kaltenegger, Andreas Leithner

**Affiliations:** Department of Orthopedics and Trauma, Medical University Graz, 8036 Graz, Austria; nicole.eck@medunigraz.at (N.E.); dietmar.glaenzer@medunigraz.at (D.G.); heike.kaltenegger@medunigraz.at (H.K.); andreas.leithner@medunigraz.at (A.L.)

**Keywords:** titanium aluminium vanadium alloy, biocompatibility, adhesion, mesenchymal stem cells, osteogenic differentiation

## Abstract

Osteogenic cells are strongly influenced in their behaviour by the surface properties of orthopaedic implant materials. Mesenchymal stem and progenitor cells (MSPCs) migrate to the bone–implant interface, adhere to the material surface, proliferate and subsequently differentiate into osteoblasts, which are responsible for the formation of the bone matrix. Five surface topographies on titanium aluminium vanadium (TiAl_6_V_4_) were engineered to investigate biocompatibility and adhesion potential of human osteoblasts and the changes in osteogenic differentiation of MSPCs. Elemental analysis of TiAl_6_V_4_ discs coated with titanium nitride (TiN), silver (Ag), roughened surface, and pure titanium (cpTi) surface was analysed using energy-dispersive X-ray spectroscopy and scanning electron microscopy. In vitro cell viability, cytotoxicity, adhesion behaviour, and osteogenic differentiation potential were measured via CellTiter-Glo, CytoTox, ELISA, Luminex^®^ technology, and RT-PCR respectively. The Ag coating reduced the growth of osteoblasts, whereas the viability of MSPCs increased significantly. The roughened and the cpTi surface improved the viability of all cell types. The additive coatings of the TiAl_6_V_4_ alloy improved the adhesion of osteoblasts and MSPCs. With regard to the osteogenic differentiation potential, an enhanced effect has been demonstrated, especially in the case of roughened and cpTi coatings.

## 1. Introduction

Titanium and its alloys, especially TiAl_6_V_4_, have long been used for orthopaedic implants. Materials that are defined as biocompatible can be embedded in living tissue without causing negative effects. Titanium is considered biocompatible due to its low electrical conductivity and its ability to form a thin passive oxide layer in an aqueous solution, resulting in high corrosion resistance [1]. Although TiAl_6_V_4_ has proven to be biocompatible in everyday clinical practice, efforts have continued to improve its osteoconductive and osteogenic properties to enhance implant performance. Furthermore, there are concerns about the cytotoxicity of vanadium and aluminium [2]. The successful integration as a bone substitute material into the surrounding tissue depends primarily on the surface structure and its properties. Chemical and physical processes such as polishing, sandblasting, plasma spraying, acid etching or bioactive coatings have been identified as a possible approach to improve the acceptance and adhesion of the involved cell types, leading to faster bone integration under in vivo conditions [3,4,5,6].

In addition to osteoblasts, mesenchymal stem and progenitor cells (MSPCs) in particular play an important role in these processes [7,8]. MSPCs from bone marrow and surrounding tissues are involved in matrix remodelling at the bone–implant interface [8,9]. Integrins belong to those cell receptors that play an important role in the extracellular environment, in adhesion, and in the distribution of cells on the material surface. Thus, they are responsible for the interactions between osteogenic cells and the material surfaces [10,11].

In the field of biomaterial development, many studies have independently considered the effects of surface topography and surface chemistry on the in vitro and in vivo biocompatibility of bone, but few have investigated the concurrent biological effects on different cell types. In this study, we attempted to perform a comparative analysis of five commercially available surface modifications of TiAl_6_V_4_ with regard to their biocompatibility and osteoinduction at the cellular level. The evaluation of the in vitro cytocompatibility of these surfaces, evaluated with a human osteoblastic cell line and primary human mesenchymal stem cells, is discussed as a function of the topography of the tested surfaces. Untreated TiAl_6_V_4_ discs compared to titanium nitride (TiN) coating, silver (Ag) coating, a roughened surface and a pure titanium (cpTi) coated surface was investigated regarding cell proliferation, cytotoxicity, adhesion potential, and the expression of osteogenic markers.

## 2. Materials and Methods

### 2.1. TiAl_6_V_4_ Alloy Surface Modifications

Using a precision casting process the TiAl_6_V_4_ discs used in this study were manufactured by Implantcast (Buxtehude, Germany). The special coatings used were provided by DOT Ltd. (Rostock, Germany). According to ISO 5832-3 chemical characterisation, the wrought alloy contains 5.5–6.75% aluminium (Al), 3.5–4.5% vanadium (V), and less than 1% O, N, H, Fe, and C, as well as pure titanium. The tensile strength R_m_ was >860 MPa (pull off test), the yield strength R_p0.2_ was >760 Mpa, and the elongation at break was ≥10%. The silver (Ag) coating makes an electrolytically deposited, sandblasted, silver white, satin metal layer with a coating thickness of 15 ± 5 µm. During the conventional manufacturing process, the roughened surface is made with aluminium oxide white F30 blasting (5.5 bar).

The ceramic surface coating with titanium nitride (TiN) is considered in clinical application to be anti-allergic, abrasion-reducing, and biocompatible. The TiN coating is securely anchored in the implant surface by an additive process. This is done by releasing titanium atoms from a solid target through electrical energy, ionisation and acceleration onto the implant surface. The golden-yellow ceramic TiN layer was deposited by physical vapour deposition and is 5.5 ± 1.5 µm thick, with an average surface roughness Ra < 0.05 µm. The adhesion to the substrate is one of the most important parameters for the quality of an applied layer. Various qualitative tests are carried out on the coatings to reliably assess the adhesion of the TiN coating. The Rockwell-HRC indentation test according to VDI guideline 3824 (HF 1-4) and the mandrel bending test are among these test methods. Since the manufacturer uses these methods for quality control, we evaluated the adhesion strength according to part 4 of the guideline VDI 3824. For quantity control, the coating was impressed for 30 s with a Rockwell “C” diamond tip under a load of 150 kg. The resulting indentation scar was examined by SEM for the presence of film cracks and delaminations. Adhesion strength was classified as HF1-HF4 depending on the severity of the damage, with HF1 signifying slight cracking and HF4 signifying severe delamination of the coating. The mandrel bend test is a standard test method to qualify the coating resistance to cracking, upon bending. With the mandrel bending test, an efficient adhesive strength could be given for the TiN coating (>22 MPa).

Vacuum plasma spraying was used to apply the coating with commercially pure titanium (cpTi). This resulted in a rough and porous surface layer with a porosity of 20–40%, an average surface roughness Ra of 50 ± 15 µm and a thickness of 250–350 µm. In order to be usable for the cell culture experiments in 24 well plates, all materials were produced as discs with a diameter of 14 mm and a thickness of 1 mm. Sterilisation is carried out by gamma irradiation according to a standardised protocol.

### 2.2. Scanning Electron Microscopy (SEM)

SEM studies were performed under high vacuum conditions and 20 kV high voltage using an FEI Quanta 250 FEG (Thermo Fisher Scientific, Hillsboro, OR, USA). The micrographs were recorded with the Everhart–Thornley detector in secondary electron mode. To ensure sufficient electrical conductivity, the surfaces of the material discs were sputter-coated with a 10 nm thin gold layer. Energy dispersive X-ray spectroscopy (EDX) measurements were performed using a 30 mm^2^ Octane Elect Plus silicon drift detector from EDAX Ametek, (Berwyn, NJ, USA) and APEX Standard Software V1.3.1. The following settings were used: 60 s duration, 20 kV high voltage, and a spot size of 4.5.

### 2.3. Tissue Harvest and Cell Culture

hFOB1. 19 Osteoblasts (Homo sapiens, CRL-11372TM, ATCC, Manassas, VA, USA) were grown under the following culture conditions: the medium consisted of DMEM/F12 supplemented with 10% fetal bovine serum (FBS), 1% L-glutamine, 100 units/mL penicillin and 100 μg/mL streptomycin (all GIBCO, Invitrogen, Darmstadt, Germany), the incubation temperature was 34 °C, in a humidified atmosphere with 5% CO_2_. Medium change was required every three days. As described previously, human primary MSPCs were isolated from cancellous bone tissue samples obtained during routine orthodontic procedures [12]. The study protocol was approved by the ethics committee of the Medical University of Graz (EK-Nummer 29-156 ex 16/17) and informed consent was obtained from the subjects. Ten female patients aged between 25 and 35 years were enrolled in the study for stem cell collection. The bone samples (4–6 mm long) had cortical or cortical/cancellous structures. After collection, the explant cultures were purified with phosphate-buffered saline (PBS; PAA Labor, Pasching, Austria) and transferred to culture flasks. The expanded and isolated MSPCs were cultured in α-modified minimum essential medium (α-MEM; Sigma-Aldrich, Vienna, Austria), which was serum-free supplemented with 10% human platelet lysate (HPL), 2 U/mL heparin (Biochrom AG, Berlin, Germany), 2% penicillin-streptomycin, 0.5% L-glutamine, 0.2% amphotericin B and 2.5% HEPES buffer (all GIBCO Invitrogen).

### 2.4. MSPC Characterisation

According to the criteria of the International Society for Cellular Therapy [13] multilineage differentiation analysis and characterisation using flow cytometry were performed as follows: flow cytometry analysis was measured on a FACS LSR II System (BD Bioscience, San Jose, CA, USA) equipped with a FACSDiva™ 8.0 software (BD Bioscience, San Jose, CA, USA) and evaluated with FCS Express 6 software (DeNovo Software, Los Angeles, CA, USA). The monoclonal antibodies CD73 PE, CD90 APC, CD105 PE, CD45 APC-Cy7, CD34 APC, CD14 FITC, CD19 APC, and HLA-DR APC (BD Bioscience) were used for the characterisation of the MSPC. In order to analyse the data of all donors consistently, 10,000 events were measured. For the induction of osteogenesis, the differentiation medium was supplemented with 100 nM dexamethasone, 0.1 mM ascorbic acid-2-phosphate, and 10 mM β-glycerophosphate (all Sigma Aldrich, St. Louis, MI, USA). After a 7-day and 14-day incubation period, alkaline phosphatase (ALPL) activity was measured by histochemical staining (Alkaline Phosphatase Kit No. 85; Sigma Aldrich, St. Louis, MI, USA). To quantify ALPL enzyme activity, the absorbance of the p-nitrophenol phosphate product was calculated at 405 nm. Differentiation of MSPCs into the adipogenic lineage was performed by supplementing the medium with 100 nM dexamethasone, 50 µM indomethacin (Sigma Aldrich, St. Louis, MI, USA) and 0.135 IU/mL insulin (Novo Nordisk, Bagsværd, Denmark). The resulting adipocyte-specific fat vacuoles were detected by Oil Red O staining after the 21-day differentiation period. The chondrogenic differentiation medium contains 10% FBS, 100 nM dexamethasone and 1 ng/mL TGF-β3 (Lonza, Basel, Switzerland) as supplements. After a 21-day differentiation period, the production of glycosaminoglycans and mucopolysaccharides was detected by Alcian blue staining (fixation with 10% formaldehyde and staining with 1% Alcian blue in 3% acetic acid solution pH 2.5).

### 2.5. Viability Assay

TiAl_6_V_4_ discs inserted into Corning Costar Ultra-Low Attachment Multiwell plates with 24 wells (Corning Inc., Corning, NY, USA) served as a growth surface for 2.5 × 10^4^ osteoblasts. After a 4-day incubation period, the CellTiter-Glo™ Luminescence Cell Viability Assay (Promega, Madison, MA, USA) was performed according to the manufacturer’s instructions. Culture media without cells served as a background reference value. Absorbance values were measured with the LUMIstar^®^ Omega microplate luminometer (BMC Labtech, Ortenberg, Germany).

### 2.6. Cytotoxicity Assay

Using the CytoTox-ONE™ Homogeneous Membrane Integrity Assay (Promega) the activity of lactate dehydrogenase (LDH) was measured after a 4-days incubation period from the cell culture supernatant. In brief, 50 µL working solutions were mixed with 50 µL of cell culture supernatant in white 96-well microtiter plates. After an incubation of 10 min (room temperature; darkness), the reaction was stopped by adding 50 µL stop solution. Fluorescence was measured at 560/590 nm using a FLUOstar^®^ Omega microplate reader (BMC Labtech).

### 2.7. Enzyme Immuno Assay (ELISA)

After 4 d incubation on the different TiAl_6_V_4_ material discs, undiluted cell culture supernatants were analysed using the human Fibronectin ELISA kit (Abcam, Cambridge, UK) according to the corresponding protocol. Each measurement was performed in duplicate at 450 nm using a SPECTROstar^®^ microplate reader (BMC Labtech).

### 2.8. Real-Time RT-PCR

After a 3-week differentiation period on the TiAl_6_V_4_ surfaces, total RNA was isolated from the MSPCs using the RNeasy Mini Kit and DNase I treatment was performed according to the manufacturer’s instructions (Qiagen, Hilden, Germany). Reverse transcription was performed from one μg of RNA using the iScript cDNA Synthesis Kit, (BioRad Laboratories Inc., Hercules, CA, USA). The real-time RT-PCR run was performed according to an established 3-step PCR temperature protocol with an annealing temperature of 60 °C. A melting curve protocol was used to detect primer dimerisation and confirm a single gene-specific peak. Relative quantification of expression levels was performed using the ΔΔCt method. The housekeeping genes TBP (TATA-box binding protein) and RPLP0 (ribosomal protein, lateral stalk, subunit P0) were used as reference genes. The expression levels of the target genes (Ct) were normalised to the housekeeping genes (ΔCt). The ΔΔCt value corresponds to the difference between the ΔCt value of the test sample and the ΔCt of the control sample. The following QuantiTect primer assays (Qiagen) were used for real-time RT-PCR: vinculin, the integrin subunits ITGβ, ITGα1, ITGα3, and ITGα5, alkaline phosphatase (ALPL), osteopontin (SPP), and bone morphogenetic protein 2 (BMP2).

### 2.9. xMAP Human Adhesion Magnetic Bead Panel

Using the Luminex^®^ xMAP^®^ platform, we measured the following analytics in cell culture supernatant samples: E-selectin (CD62E), P-selectin (CD62P), soluble intercellular adhesion molecule-1 (sICAM-1), soluble vascular adhesion molecule-1 (sVCAM-1), plasminogen activator inhibitor-*1* (PAI-1), and Platelet endothelial cell adhesion molecule (PECAM; CD31). Cross-reactivity between the antibodies could be excluded.

### 2.10. Statistical Analysis

Statistical differences between groups were analysed using the unpaired Student’s t-test and the Wilcoxon exact test (SigmaPlot 14.0; Systat Software Inc., San Jose, CA, USA). The following values were considered statistically significant: *** *p* < 0.001; ** *p* < 0.01; * *p* < 0.05. Graphical representations were also designed with SigmaPlot 14.0.

## 3. Results

### 3.1. Surface Characteristics

As can be clearly seen in the SEM images, there are significant differences in the morphology of the uncoated TiAl_6_V_4_ discs and their modifications (1000- and 5000-fold magnification shown as inserts) (Figure 1). The TiN coating bonded extremely well to the implant and macroscopically showed a metallic, golden-yellow appearance. When SEM magnified, one can see the slightly roughened surface of this coating. The coating with silver (Ag) resulted in a surface with very few structures. In contrast, the roughened surface shows significantly more structure than the original alloy. The coating with commercial pure titanium (cpTi) produced a structured surface with rounded elements, which is created by the special coating process of vacuum plasma spraying. EDX analysis revealed changes of the surface quality between the uncoated TiAl_6_V_4_ surface and the modifications regarding the composition of the chemical elements (Figure 1).

### 3.2. MSPC Characterisation and Multilineage Differentiation Analysis

Cells were isolated from all samples within 8 d, which exhibited the mononuclear, fibroblast-like, spindle-shaped, plastic-adherent characteristics of human primary MSPCs. Our primary human MSPCs showed positive expression of CD73 (99.8 ± 0.1%), CD90 (99.8 ± 0.2%), CD105 (69.1 ± 9.8%) of gated cells in flow cytometric analyses. The typical forward/side scatter characteristics of 71.5 ± 4.9% were gated. Furthermore, the negative expression of CD14 (0.2 ± 0.2%), CD19 (0.6 ± 0.1%), CD34 (0.4 ± 0.3%), CD45 (23.9 ± 7.8%) and HLA-DR (0.5 ± 0.3%) confirms the typical phenotype of MSPCs (Figure 2a). This is in accordance with the criteria of the International Society for Cellular Therapy [13] for defining multipotent mesenchymal stromal cells. Furthermore, the cells were successfully differentiated towards the osteogenic, chondrogenic and adipogenic lineages, demonstrating the pluripotent properties of the MSPCs. ALPL activity was measured of absorbance (optical dense, OD) of p-nitrophenol in supernatant at 405 nm over 14 d (Figure 2b). ALPL expression showed a significant increase (*p* < 0.001) in osteogenically differentiated MSPCs at 7 d and 14 d and over time. No ALPL was measured in the undifferentiated negative controls. Based on the interaction of the dye Alcian blue with glycosaminoglycans, chondrogenically differentiated MSPCs showed increased blue staining compared to undifferentiated controls. Aggrecan expression increased 4.7-fold (*p* < 0.05) as a result of chondrogenic differentiation (Figure 2c). To prove the pluripotency of the MSPCs, cells were also differentiated in the adipogenic line. As a result of the 21-day adipogenic differentiation, lipid vacuoles formed and could be visualised by oil red O staining (Figure 2d). The sum of the characterisation experiment clearly identifies our primary cells as MSPCs.

### 3.3. In Vitro Biocompatibility Assays

The CellTiter-Glo^®^ viability test (Figure 3a) and the CytoTox-ONE™ cytotoxicity test (Figure 3b) were performed to analyse the biocompatibility of the different surface modifications after an incubation period of 4 d. Human osteoblasts (hFOB) and primary MSPCs showed an even distribution on all surfaces. To compare the differences between the modified surfaces, the original unmodified TiAl6V4 was used as a reference (100%). Data were shown as mean ± SD; *n* = 8; measured in quadruplicate. The ceramic TiN coating showed few differences to the uncoated TiAl_6_V_4_ alloy in the different cell types. The silver coating reduced the growth of osteoblasts 67.8 ± 14.7% (*** *p* = 4.9 × 10^−12^), whereas the viability of both undifferentiated and osteogenically differentiated (OG) MSPCs increased significantly (undiff.: 134.2 ± 4.5%; *** *p* = 8.7 × 10^−23^; OG: 110.8 ± 3.9%; *** *p* = 8.6 × 10^−10^). The roughened and the pure titanium (cpTi) surface improved the viability of all cell types (Figure 3a).

The amount of fluorescence is proportional to the number of lysed cells because lactate dehydrogenase (LDH) is released from the cells when the membranes of necrotic cells are disturbed. No change in LDH release and associated cytotoxicity was observed between the groups (Figure 3b). Only the silver coating showed reduced values (mean ± SD; *n* = 6, measured in quadruplicates).

### 3.4. Expression of Cellular Adhesion Proteins on Different TiAl_6_V_4_ Surface Modifications

We analysed the expression of adhesion markers using 6plex Luminex^®^ technology. Data were presented as concentrations calculated from fluorescence intensity minus background using 5-parameter logistic regression (mean ± SD; *n* = 8, measured in duplicates) (Figure 4). The expression of PECAM and P-selectin were below the detection limit. E-selectin revealed no significant differences between the surfaces (Figure 4a). The coating with pure titanium (cpTi) caused a changed expression of PAI-1 (TiAl_6_V_4_: 7698 ± 223 vs. cpTi: 7440 ± 157; * *p* = 0.01) (Figure 4b) and sVCAM-1 (TiAl_6_V_4_: 172 ± 53 vs. cpTi: 220 ± 29; * *p* = 0.03) (Figure 4c). sICAM expression was significantly increased by the Ag coating (TiAl_6_V_4_: 467 ± 72 vs. Ag: 568 ± 44; * *p* = 0.02) (Figure 4d).

Fibronectin, a glycoprotein of the extracellular matrix, binds to integrin receptor proteins associated with tissue remodelling and repair. It was observed that there was no significant difference in fibronectin synthesis by cells seeded on all different material groups as compared to TiAl_6_V_4_ control group (Figure 5a). The membrane–cytoskeleton interactions were analysed using the gene expression analysis of vinculin. The ceramic TiN coating (relative mRNA levels: 2.05 ± 0.28; *** *p* = 0.0009), the roughened surface (1.90 ± 0.66; * *p* = 0.03), and the cpTi coating (2.30 ± 0.37; ** *p* = 0.001) revealed a significant increase in vinculin expression compared to the TiAl_6_V_4_ control group, whereas the Ag overlay reduced the formation of vinculin (0.56 ± 0.36; * *p* = 0.02) (Figure 5b).

Since ITG-α3β1 and ITG-α5β1 play important roles as fibronectin receptors in osteoblasts, we examined the expression of integrin subunits α1, α3, α5 and β1 by RT-PCR after a 4-day incubation period. The untreated TiAl6V4 alloy served as the reference value (ratio = 1; mean ± SD; experiments performed in triplicates) (*n* = 8) (Figure 5c–f). Osteoblasts cultivated on TiN coated material revealed a highly significant increased expression of ITG-β1 (2.22 ± 0.42; *** *p* = 0.0007) and ITG-α1 (2.36 ± 0.28; *** *p* = 4.37 × 10^−5^). Ag coating resulted in an increase in ITG-β1 (1.57 ± 0.27; ** *p* = 0.003) and ITG-α1 (2.93 ± 0.47; ** *p* = 0.0001). The roughened surface showed no significant changes. An increased expression of integrin subunits can also be observed on the cpTi surface (ITG-β1: 2.09 ± 0.21; ** *p* = 1.83 × 10^−5^ and ITG-α1: 1.35 ± 0.19; ** *p* = 0.005).

### 3.5. Influence of TiAl_6_V_4_ Surface Modifications on Osteogenic Differentiation of MSPCs

An important aspect of the integration of the prosthesis into the surrounding tissue is the osteogenesis of the immigrated MSPCs. Undifferentiated primary MSPCs were osteogenically differentiated onto the different TiAl_6_V_4_ discs for three weeks. After RNA isolation RT-PCR was performed for detection of the relative expression of ALPL, osteopontin (SPP), and BMP2 (mean ± SD; *n* = 9, measured in triplicates) (Figure 6).

ALPL expression increased significantly by 2 to 5-fold in all groups due to osteogenic differentiation.The cpTi coating effected a particularly significant increase to 5.21 ± 1.24; *** *p* = 0.0002. In osteogenically differentiated groups, a 7- to 10-fold higher SPP expression was observed. Again, for this osteogenic marker, the cpTi coating showed the best performance (12.47 ± 1.32; ** *p* = 0.004). The BMP2 expression increased significantly in all surface modifications as a result of osteogenic differentiation. Interestingly, however, the cpTi coating reduced this highly significant to 0.29 ± 0.09; *** *p* = 2.85 × 10^−7^ (Figure 6a). If the osteogenic differentiation potential was put into relation to the untreated TiAl_6_V_4_ alloy, the ALPL expression showed a 2-fold higher potential for the roughened surface and a 4-fold increase for cpTi (Figure 6b). cpTi coating increased SPP 2-fold. The roughened surface showed a 3-fold increased expression of BMP2 in relation to the untreated TiAl_6_V_4_ alloy, whereas with cpTi it decreased to a tenth.

## 4. Discussion

TiAl_6_V_4_ alloys show high specific strength and are considered biocompatible due to their low electrical conductivity. This oxide layer in turn leads to high corrosion resistance [14,15]. Nanostructured metal-based layers are used to increase the biocompatibility, adhesion, growth and phenotypic maturation of osteoblasts. One intensively researched area is modifications of the material surfaces in terms of roughness and specific coatings. This should increase the attractiveness of the biomaterial for cells and improve integration into the surrounding tissue [16,17]. The majority of these studies concluded that surface structures on the nanoscale interact with cell adhesion molecules, resulting in better adhesion of the cell types involved and thus supporting proliferation.

In our study, we investigated the biocompatibility and the adhesion potential of human osteoblasts and primary MSPCs seeded on different surface modified TiAl_6_V_4_ alloy, i.e., titanium nitride (TiN) coating, silver (Ag) coating, roughened surface, and a coating with pure titanium (cpTi). Exactly these surfaces are also used in orthopaedic surgery. To ensure good comparability, all the materials we used in this study were produced by a commercial manufacturer of orthopaedic implants. Our biocompatibility experiments showed a very good acceptance of the cpTi and Ag coatings on osteoblasts as well as in undifferentiated and osteogenic differentiated MSPCs. The measured values are significantly above those of the untreated TiAl_6_V_4_ alloy. The improved biocompatibility of cpTi can be explained by a higher corrosion resistance and a stable and inert oxide layer [18]. Silver coatings are primarily used in medical implants as an antibacterial agent to reduce biofilm formation. The damage to the bacterial membranes is caused by the direct contact with silver nanoparticles, which leads to a change in the cell membrane proteins [19,20]. The effect of silver nanoparticles on the adhesion and proliferation of fibroblasts showed very different properties of biointegration [21,22]. Yuan et al. confirmed that an Ag multilayer film has negligible cytotoxicity and promotes osteoblast growth in comparison, which could be attributed to the slow release of Ag ions [23]. In addition, none of the investigated materials showed cytotoxic effects on osteoblasts and MSPCs.

There is a direct correlation between the cell adhesion and the interaction of hard and soft tissue cells with the implant surface to the tissue integration of the implant [24]. To investigate these cell–cell and cell-matrix interactions, we analysed the expression of the extracellular matrix protein fibronectin, the membrane cytoskeleton protein vinculin, which is involved in linking integrins to the actin cytoskeleton, and various integrin receptor subunits. The TiAl_6_V_4_ surface modifications we examined did not alter fibronectin expression cells. In contrast, TiN, the roughened surface and cpTi significantly enhanced the expression of vinculin. The Ag coating on the other hand reduced the expression significantly. Whether these changes are due to the chemistry of the surface coatings or to the different degrees of roughness cannot be determined exactly from our data.

The adhesion of cells to the extracellular matrix proteins is primarily mediated by the integrin receptor family. [25]. Integrins of the subfamily β1 are the predominant mediators of cell adhesion and osteoblastic differentiation in osteoblasts and osteoprogenitors [26]. Some studies have identified combinations of α1β1, α2β1, α3β1, α5β1, αvβ3 integrins in osteoblasts and bone cultures [27,28,29]. Regarding the expression of the integrin subunits we could observe a significantly increased ITG-α1 and ITG-β1 expression in all TiAl_6_V_4_ surface modifications. Especially those surfaces with low structural elements, like TiN and Ag, showed a strong increase. These findings indicated that integrin α1β1 signalling is required for the adhesion potential on TiAl_6_V_4_ microstructures. During osteoblast differentiation on microstructured titanium substrates, specific integrin-receptor complexes also regulate angiogenic factor production [30].

In order to improve the biocompatibility of TiAl_6_V_4_ alloys, there have always been efforts to improve the osteoconductive and osteogenic properties to enhance the performance of the implant [31,32,33]. MSPCs from the bone marrow and surrounding tissue migrate to the bone–implant interface after implantation of an orthopaedic prosthesis and participate in matrix remodelling. Thus the osteogenic differentiation potential of the MSPCs plays a decisive role in the healing process. Primary human MSPCs were isolated from cancellous bone and characterised according to the criteria. We measured the level of osteogenic differentiation markers by gene expression analysis. Unfortunately, RNA isolation of cells grown on Ag discs wasn’t possible. This phenomenon may be due to interactions of components of the osteogenic differentiation medium with the silver surface. After the 3-week differentiation phase, the MSPCs seeded on Ag discs showed a changed morphology and partially floated off. Therefore, this group is missing in the analyses. Alkaline phosphatase (ALPL), an early indicator of osteogenic differentiation, increased with progressive duration [34]. The osteogenic differentiation caused the values of ALPL depending on the respective coating to increase 2–5-fold. The highest increase could be observed with the cpTi coating. Osteopontin (SPP) is a secreted adhesive glycophosphoprotein and plays a key function within bone tissue in cell adhesion, migration and survival [35]. The osteogenic differentiation caused a highly significant 7–10-fold increase in SPP. Compared to the TiAl_6_V_4_ control group, the cpTi coating also showed the best performance with this osteogenic differentiation marker. Bone morphogenetic protein (BMP) ligands are upregulated temporally during MSPCs differentiation on microstructured titanium substrates [36]. BMP2 plays an important role in embryonic bone formation and adult skeletal homeostasis [37]. Compared to the TiAl_6_V_4_ control group, the cpTi group revealed a highly significant decrease, while TiN and the roughened surface showed an increase in BMP2 expression. These findings are in agreement with the results of Olivares-Navarrete et al. in which increased roughness of the TiAl_6_V_4_ surface led to significantly increased mRNA levels of BMP2 and BMP4 in human MG63 osteoblast-like cells [38].

## 5. Conclusions

Although all TiAl_6_V_4_ surface modifications investigated in this study are used in daily orthopaedic surgery, no direct comparison study exists to date. The present data on the behaviour of individual cell types are an important contribution to the understanding of the cell biological processes at the prosthesis–tissue contact sites. The additive coatings of the TiAl_6_V_4_ alloy improve the proliferation and adhesion of human osteoblasts and primary MSPCs. Especially those materials with more structured surfaces, such as the roughened and cpTi coatings, further increased the osteogenic differentiation potential of MSPCs.

## Figures and Tables

**Figure 1 materials-14-01574-f001:**
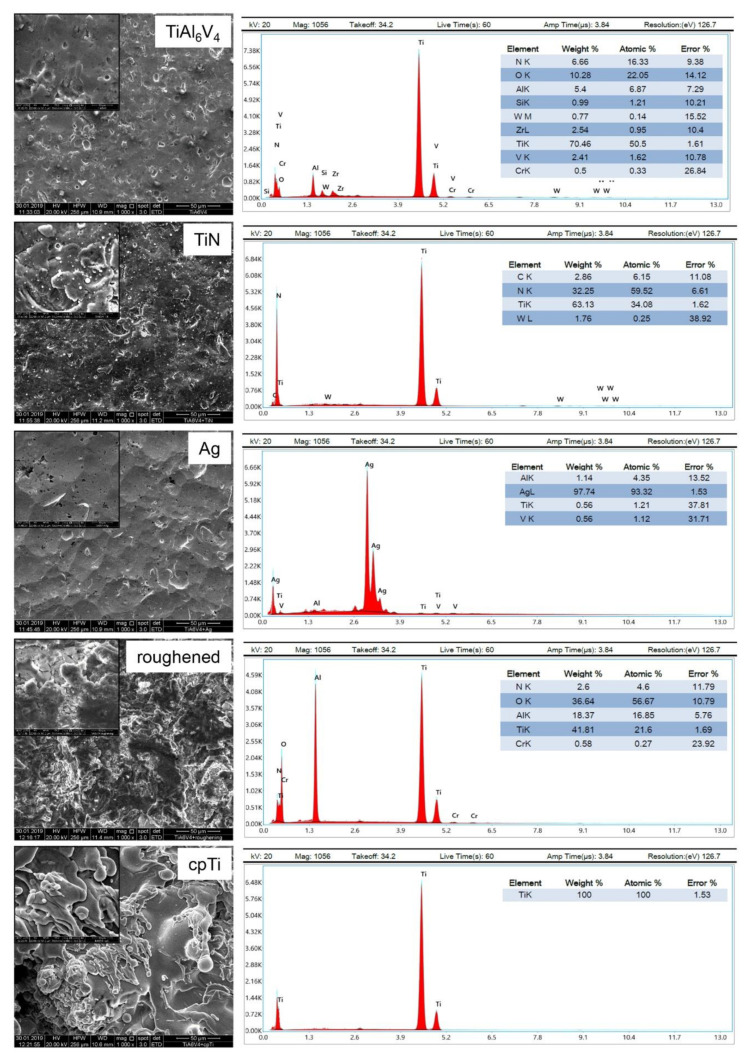
Surface characteristics. SEM (Scanning Electron Microscopy) energy dispersive X-ray spectroscopy (EDX) analysis of uncoated TiAl_6_V_4_, titanium nitride (TiN) coated, silver coated (Ag), a roughened surface, and a pure titanium coated surface (cpTi) (all 1000× and 5000× magnification). The corresponding EDX analysis exhibited changes of the surface quality between the uncoated TiAl_6_V_4_ surface and the modifications regarding the composition of the chemical elements.

**Figure 2 materials-14-01574-f002:**
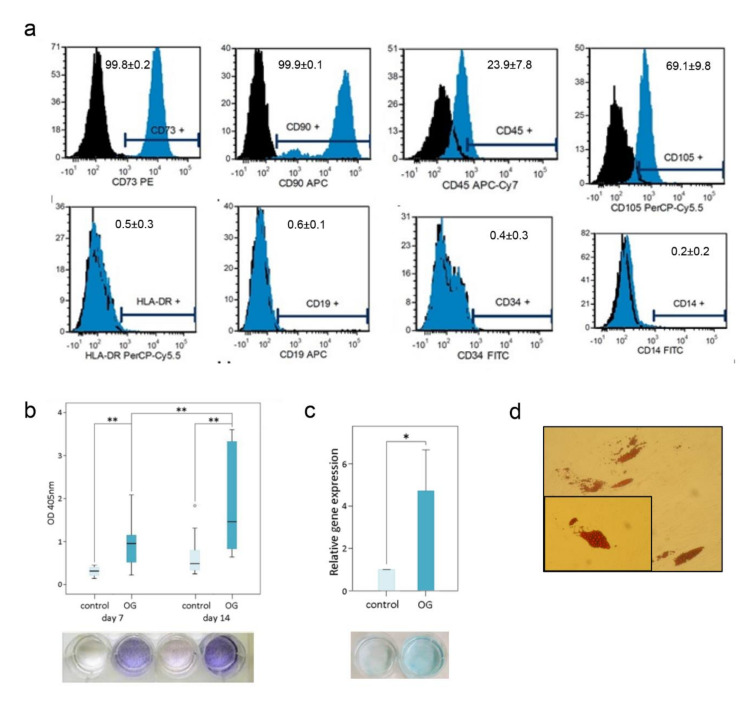
Multilineage differentiation analysis and flow cytometry surface characterisation of mesenchymal stem and progenitor cells (MSPCs). The isolated MSPCs were characterised according to (**a**) the positive expression of CD73, CD90, CD105 and the negative expression of CD14, CD19, CD34, CD45 and HLA-DR by flow cytometry (black: unstained cells; blue: stained cells). The differentiation potential was clearly demonstrated by (**b**) alkaline phosphatase (ALPL), osteocalcin and osteopontin expression for osteogenic differentiation, (**c**) Alcian blue staining and the expression of aggrecan for chondrogenic differentiation and (**d**) Oil Red O staining of lipid droplets for the adipogenic lineage. Statistical significances are defined as follows: ** *p* < 0.01; * *p* < 0.05.

**Figure 3 materials-14-01574-f003:**
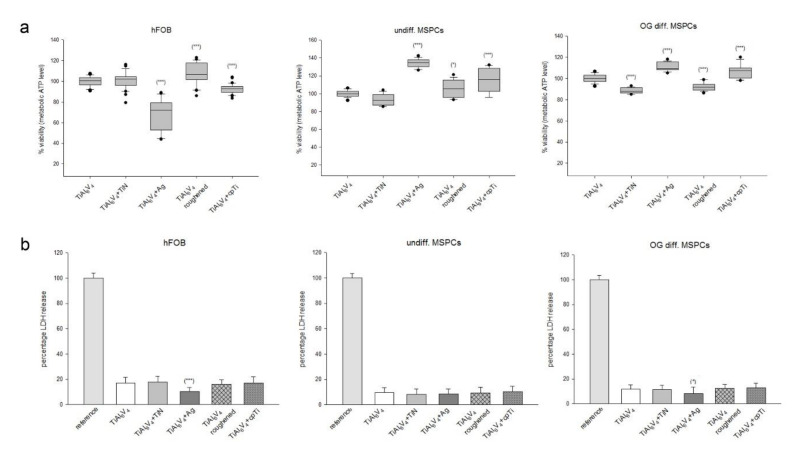
The influence of TiAl_6_V_4_ surface modifications on viability and cytotoxicity. (**a**) The cell viability of human osteoblasts, undifferentiated MSPCs, and osteogenic differentiated (OG) MSPCs on uncoated TiAl_6_V_4_, titanium nitride (TiN) coated, silver coated (Ag), a roughened surface, and a pure titanium coated (cpTi) surface was represented with box plots (mean ± SD; *n* = 8; measured in quadruplicate). Statistical significances are defined as follows: *** *p* < 0.001; ** *p* < 0.01; * *p* < 0.05. (**b**) No significant changes were seen in the evaluation of cytotoxicity (mean ± SD; *n* = 6; measured in quadruplicate).

**Figure 4 materials-14-01574-f004:**
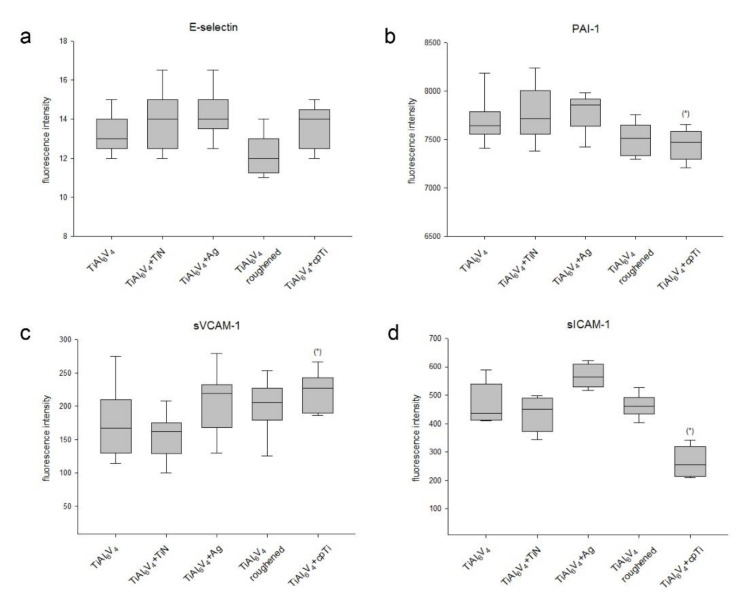
Expression of adhesion markers. The expression of six adhesion markers was analysed with the Luminex^®^ xMAP^®^ platform on uncoated TiAl_6_V_4_, titanium nitride (TiN) coated, silver coated (Ag), a roughened surface, and a pure titanium coated (cpTi) surface. The expression of (**a**) E-selectin, (**b**) PAI-1, (**c**) sVCAM-1, and (**d**) sICAM-1 was presented in box plots (mean ± SD; *n* = 8, measured in duplicates). P-selectin and PECAM were out of range. Statistical significance is defined as follows: * *p* < 0.05.

**Figure 5 materials-14-01574-f005:**
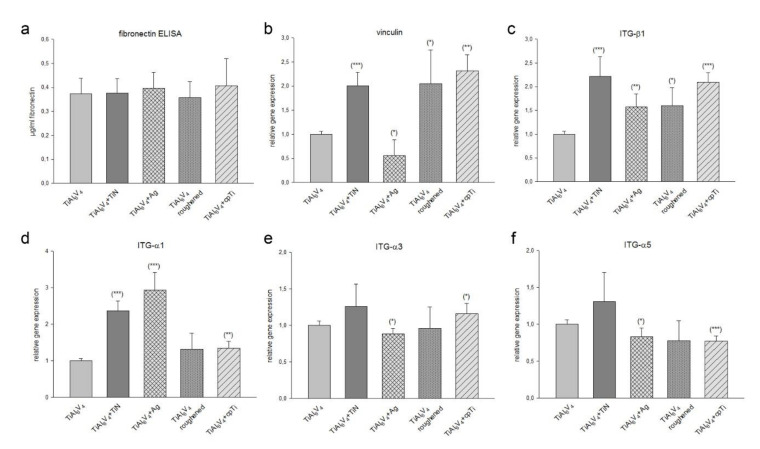
The influence of TiAl_6_V_4_ surface modifications on the expression of adhesion molecules. (**a**) Fibronectin expression showed no changes between the groups. The relative gene expression of (**b**) vinculin, and (**c**) the integrin subunits ITG-β1, and (**d**–**f**) ITG-α1, -α3, α5 was presented (mean ± SD; *n* = 8; measured in triplicates). Statistical significances are defined as follows: *** *p* < 0.001; ** *p* < 0.01; * *p* < 0.05.

**Figure 6 materials-14-01574-f006:**
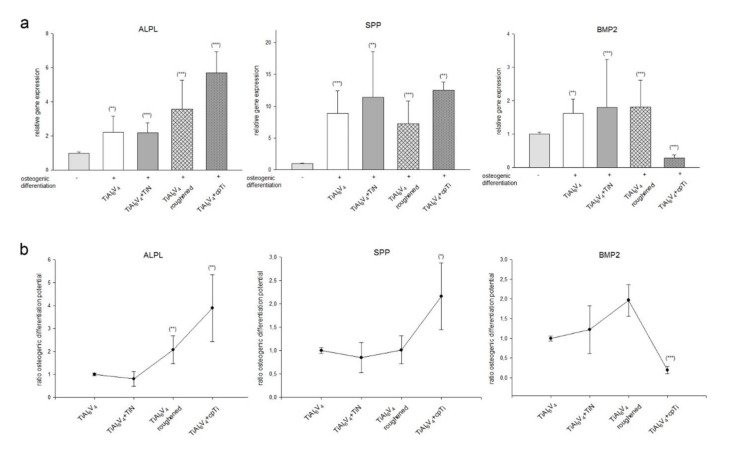
The expression of osteogenic markers in MSPCs seeded on different surface modified TiAl_6_V_4_ discs. MSPCs were seeded on uncoated TiAl_6_V_4_, titanium nitride (TiN) coated, silver coated (Ag), a roughened surface, and pure titanium coated material discs and the cells were induced for an osteogenic differentiation over 21 d. After RNA isolation the relative expression of ALPL, SPP, and BMP2 were analysed using RT-qPCR. Undifferentiated MSPCs (-) served as references (ratio = 1) (mean ± SD; *n* = 9, measured in triplicates). (**a**) shows the change in the expression of important osteogenic differentiation markers on the TiAl_6_V_4_ surfaces, (**b**) displayed the changes within the material groups in relation to the untreated TiAl_6_V_4_ group. Statistical significances are defined as follows: *** *p* < 0.001; ** *p* < 0.01; * *p* < 0.05.

## Data Availability

All data obtained are included in the study.

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
