# Peer review of "Surface Modifications of Titanium Aluminium Vanadium Improve Biocompatibility and Osteogenic Differentiation Potential"

_materials, 2021, doi:10.3390/ma14061574_

Round 1

Reviewer 1 Report

The reviewed paper presents results investigation on the compare of five commercially available surface modifications of TiAl6V4with regard to their biocompatibility  and osseoinduction at the  cellular level. The received results exhibited  that a modification of the surface structure of implants is sufficient to create  an osteogenic environment that could improve bone  formation and implant stability.

The paper requires the following amendments:

Absract:

line 18; what does it mean "roughened surface" -  whether it is one of the roughness-modified studied coatings or the Ti6Al4V substrate, with a mechanically modified surface?

Introduction:

Lines: 35-39; this is a very general statement that should be expanded in this paper section. 
What is the novelty of the conducted research compared to the previous reports? 

Materials and Methods:

2.1.; it should be precisely defined what constitutes "a rouhened surface".
2.2.; in my opinion the surfaces of studied samples have not been sufficiently characterized. The surfaces of all studied coatings should be also characterized using XRD methods, surface morphology should be determined also by AFM method, moreover the contact angle should be determined, and their mechanical properties. In the case of Ag coating, the processes of Ag+ ion release in the environment of body fluids should also be investigated.

Results:

The surfaces of studied samples have not been sufficiently characterized. The physicochemical properties, wettability and mechanical properties of the surfaces of all tested materials should be compared.
3.5 and Figure 6: an explanation why there is no data on the Ti6Al4V/Ag system should be found in this part of the paper.

Discussion:

In this part of the article, the authors should discuss the differences in surface properties of the tested materials and their influence on biological activity.

In my opinion, the reviewed paper requires the major alterations before being recommended for the publication in Materials. 

Author Response

Thank you for your efforts to improve our manuscript with suggested corrections. 
Please find enclosed our comments and corrections.

Reviewer 2 Report

The aim of the manuscript “Surface modifications of titanium-aluminium-vanadium improve biocompatibility and osteogenic differentiation potential” was to comparatively analyze biocompatibility and osseoinduction, at cellular level, five surface modifications of TiAl6V4 alloy commercially available for orthopedic purpose.

The presented study is well conducted, modern and very significant in the field. The manuscript is well structured and the research is very useful for orthopedic application as well as for maxillofacial reconstructions.

Some minor issues need to be addressed.

Introduction

“Integrins, which act as important cell receptors for the extracellular environment and play a central role in the adhesion and distribution of cells to material surfaces, thus play an important role in the interactions between osteogenic cells and material surfaces” (rows 46-48) -  need to be rephrased.

Material and Methods

Please describe with more details the tests performed for the adhesion to the substrate of TiN coating, respectively Rockwell HRC indentation test and mandrel bending test (HF 1-4) and mandrel bending test > 22 MPa (rows 74-77). Please specify the significance of the tests and the significance of the obtained values.

Figure 4 legend – please replace “box blots” with “box plots” (row 277).

Author Response

(The authors gave the same response as above.)

Reviewer 3 Report

The authors provided different surface treatments on the Ti alloy for the promoted osteogenic effects as bone implants. The following problems should be addressed before publication:

1. The surface chemistry and roughness could both affect the cell behaviors, and the first three groups have totally different surface roughness when compared to roughed or cp Ti groups, so what is the main reason on the cell differentiation behaviors? The roles of different surface coating (chemistry) and roughness should be separately discussed.

2. Only in vitro data is shown in this study and it is not appropriate to conclude that "a modification of the surface structure of implants is sufficient to create an osteogenic environment that could improve bone formation and implant stability. "

Author Response

(The authors gave the same response as above.)

Round 2

Reviewer 1 Report

I propose to accept the corrected version of this manuscript for publication.